Integrating lightweight YOLOv5s and facial 3D keypoints for enhanced fatigued-driving detection

Arava Mohan
Sundaram Divya Meena divyameena.s@vitap.ac.in
SCOPE, VIT-AP University , Amaravathi, AP , India
Cirillo Stefano
Electronic publication date: 2024 Dec 5
Publication date: 2024
Volume: 10
Electronic Location ID: e2447
Received 2024 May 7; Accepted 2024 Oct 3
Copyright: © 2024 Arava and Sundaram
Copyright year: 2024
Copyright holder: Arava and Sundaram
License: This is an open access article distributed under the terms of the Creative Commons Attribution License, which permits unrestricted use, distribution, reproduction and adaptation in any medium and for any purpose provided that it is properly attributed. For attribution, the original author(s), title, publication source (PeerJ Computer Science) and either DOI or URL of the article must be cited.
License URL: https://creativecommons.org/licenses/by/4.0/

Keywords: Driver drowsiness, Improved YOLOv5s, Attention Mesh, 3D keypoint, SWIN Transformer

Funding: VIT-AP University This work was funded by a seed grant from VIT-AP University. The funders had no role in study design, data collection and analysis, decision to publish, or preparation of the manuscript.

==============================
Several factors cause vehicle accidents during driving, such as driver negligence, drowsiness, and fatigue. These accidents can be prevented if drivers receive timely warnings. Additionally, recent advancements in computer vision and artificial intelligence (AI) have enabled the monitoring of drivers and the ability to alert them when they are not focused on driving. AI techniques can analyse key facial features, such as eye closure, yawning, and head movements, to assess the driver’s level of sleepiness. In response to the growing concerns surrounding drowsy driving and its potential safety hazards, this study presents a comprehensive approach for detecting a driver’s attention state using an enhanced version of the You Only Look Once (YOLOv5) algorithm. By leveraging critical facial landmarks and calculating the eye and mouth aspect ratios, the method effectively identifies signs of fatigue by establishing threshold values indicative of closed eyes and yawning. This work introduces an advanced YOLOv5 model integrated with Swin Transformer modules in the feature fusion network and refined backbone network feature extraction to detect driver drowsiness. Additionally, a real-time fatigued-driving detection model, built on an improved YOLOv5s architecture and incorporating Attention Mesh 3D key points, demonstrates superior effectiveness over conventional models. The proposed method achieves a notable 2.4% enhancement in mean average precision (mAP) compared to the baseline model through extensive experimentation on benchmark datasets. By combining YOLOv5 with facial 3D landmarks, the system benefits from the complementary strengths of both techniques, leading to more accurate and robust detection of fatigue-related cues and ultimately mitigating accidents caused by drowsy driving.

Introduction

Drowsiness detection, a key application of human activity recognition, addresses the transition between wakefulness and sleep that slows reaction times. Fatigue, drowsiness, and driver negligence are leading causes of fatal road accidents, impairing attention and vehicle control. Factors like work stress, irregular hours, childcare, and certain medications contribute to drowsiness. The effects of drowsy driving include reduced focus, delayed reactions, poor judgment, and the risk of falling asleep while driving. With the rapid evolution of industrial technology, the landscape of transportation infrastructure has undergone fundamental shifts. While the widespread adoption of automobiles has undeniably enhanced travel efficiency and convenience, it has also introduced a heightened risk of traffic accidents (Shahrier et al., 2024). Analysis of data statistics indicates that the main reasons behind these accidents are intricately associated with factors such as exhaustion, driving under the influence, excessive cargo, and exceeding speed limits. Fatigued driving is a major contributor to traffic accidents, accounting for 14–20% of all accidents and an alarming 43% of serious accidents (Mofolasayo, 2024). Truck-related accidents on highways also account for approximately 37% of these incidents. This prevalence can be attributed to the relaxation and fatigue drivers experience after prolonged intense driving, resulting in diminished reaction times and anticipation abilities (Zafar et al., 2024). Consequently, this poses a grave threat to both life and property safety. Therefore, to reduce the frequency of traffic accidents and ensure the safety of both humans and property, it is crucial to do a thorough study on identifying fatigued driving (Das et al., 2024).

Currently, researchers are looking at ways to detect driver fatigue, especially for drivers on the road. There are three main approaches: how the vehicle is being driven, the driver’s body signals, and analysing the driver’s face with a camera (Beles et al., 2024). The most common way is to use a camera to track the driver’s facial features and head movements. Special computer programs (deep learning algorithms) can then analyse things like how often the driver blinks, the shape of their eyes, how long their eyes are closed, how wide their mouth is open, the position of their head, and even their expressions. This allows researchers to determine how tired the driver is by combining information from one or more of these features. It’s a good method because it doesn’t require any special equipment worn by the driver, and it can give accurate results and warn the driver if they get too tired (Coyne et al., 2024). Out of all the facial features that can be tracked, eye closure is especially important for detecting fatigue. In 1998, scientists developed a key measure called Percentage of Eyelid Closure over the Pupil over Time (PERCLOS), which tracked how often a driver’s eyes are closed. This has become a widely used tool for identifying fatigue. Another study by Dziuda et al. (2021) looked at professional truck drivers and confirmed that PERCLOS is a good way to tell if a driver is tired. They measured PERCLOS, how long the drivers’ eyes were closed, and how often they blinked (Perrotte et al., 2024). Their findings showed that PERCLOS was a very important measure for predicting fatigue, making it a key factor in research on detecting tired drivers.

In the early days of fatigue detection, researchers mainly focused on looking at just one facial feature at a time. For example, some studies looked at yawning specifically. Arava & Sundaram (2024) created a system to find yawns using mouth detection with an accuracy of almost 98%. Zhang & Su (2017) used a combination of computer programs to detect continuous yawning with over 87% accuracy. Knapik & Cyganek (2019) even tried using a special camera that could see heat to detect yawns, which worked in all lighting conditions. However, these methods needed fixing. They might miss important information by only looking at one feature. They could also mistake other things for yawns, especially if the driver were wearing something over their mouth or moving their head a lot. These early methods also could have been more reliable.

Researchers are now looking at combining information from multiple facial features instead of just one. This helps to overcome the problems of earlier methods, such as missing important information or mistaking other things for signs of fatigue. One popular approach is to use special computer programs (Multi-task Cascaded Convolutional Neural Network (MTCNN) and Dlib) to find important points on the driver’s face, such as around the eyes, mouth, and head. Several studies have used MTCNN to find these key points. Deng et al. (2019) used it with another program to track the eyes and head position and see how often the driver blinked. Liu et al. (2021) used MTCNN to find five key points and then combined information about the eyes, mouth, and blinking (PERCLOS) with a fuzzy logic system to decide how tired the driver was. Liu, Peng & Hu (2020) used a different method to find 24 key points and again used information about the eyes, mouth, blinking, and yawning to decide on fatigue level. However, these methods only looked at a few points on the face, which could be problematic if the driver moved their head around a lot. To address this, some researchers have turned to Dlib, which can find 68 key points on the face. This allows for a more complete picture of the driver's facial expressions, including the eye-mouth ratio and head pose. Studies by Zhang, Zhang & Huang (2019), Li, Gong & Ren (2020), Babu, Nair & Sreekumar (2022), Cai et al. (2023) all used Dlib with some success. However, there are still challenges. Dlib can only keep track of some of the points if the driver moves their head a lot, and it can be computationally expensive to run on devices inside vehicles. Overall, researchers are making progress in fatigue detection by looking at multiple facial features. However, there is room for improvement, especially when dealing with large head movements and making the systems run efficiently on low-power devices.

The key to catching drowsy drivers with cameras is tracking their facial features accurately. However, much research has focused on making the detection precise without considering how much computer power it takes to run everything. Current methods, like MTCNN’s limited key points and Dlib’s 2D approach, also have drawbacks. They are not always reliable, fast enough, or able to keep track of the driver’s face in real time. This makes it hard to put these systems in cars. So, the big challenge now is to develop systems that can still track many facial features accurately but are lightweight and work well in real-time on devices that do not have much computing power. Because of this, researchers will be able to create fatigue monitoring systems that vehicles can utilise.

Attention detection methods primarily fall into two categories: those relying on wearable devices and those leveraging machine vision. Wearable device-based methods involve collecting physiological signals like electroencephalograms, electrocardiograms, and electromyograms, offering high accuracy in reflecting the physical state of individuals. However, their limited applicability, complex operation, and high cost restrict their practicality. Conversely, machine vision-based methods analyse facial features to gauge attention diversion, offering higher real-time performance and cost-effectiveness. These methods have gained prominence with advancements in machine vision and deep learning. YOLOv3 models have been proposed for various applications in object detection, enhancing detection performance through multi-scale fusion and structural optimisations. In face detection, fatigue detection algorithms based on facial multi-feature fusion have been introduced, achieving high accuracy in detecting fatigued behaviour. Additionally, methods combining facial feature detection systems with fatigue judgment algorithms have demonstrated improved detection speed and accuracy. Furthermore, systems incorporating 3D convolutional neural networks with attention mechanisms have shown enhanced fatigue detection performance. However, most of these methods focus on static detection, excluding continuous dynamic behaviour from their detection scope. Table 1 presents the comparison of various state-of-the-art techniques in driver drowsiness detection system.

Table 1 Comparison of various state-of-the-art techniques on various features and performance.

Reference	Input parameters	Classification methods	Fatigue features	Other features	Accuracy	
Pandey et al. (2022)	Eye and face	CNN	–	–	98.53%	
Bajaj et al. (2022)	Eye and mouth	Dlib’s haar cascade model.	PERCLOS	–	98%	
Saurav et al. (2022)	Eye and face	Dual CNN Ensemble	–	–	98.98%	
Magán et al. (2022)	Face	RNN and CNN	–	–	60%	
de Lima Medeiros et al. (2022)	Eye, head, and
mouth	CNN and SVM	Eye blink		97.44%	
Moujahid et al. (2021)	Eye, head, and
mouth	SVM	–	Nodding	79.84%	
Dua et al. (2021)	Eye and face	Deep-CNN-based
ensemble	–	Nodding	85%	
Savaş & Becerikli (2021)	Mouth	ConNN model	Yawn	–	99.35%	
Maior et al. (2020)	Eye	Multilayer perceptron, RF, and SVM	PERCLOS	Facial landmarks	94.9%	
Hashemi, Mirrashid & Beheshti Shirazi (2020)	Eye	TL-VGG16, and TL-VGG19	PERCLOS	–	95.45%	

This study introduces an advanced approach to detecting driver fatigue using an enhanced YOLOv5 algorithm integrated with Swin Transformer modules. The method effectively identifies signs of driver fatigue by leveraging critical facial landmarks, such as the eye aspect ratio and mouth aspect ratio. This integration results in more accurate and robust detection of fatigue-related cues, crucial for mitigating accidents caused by drowsy driving. Furthermore, the research addresses limitations in current fatigue detection methods, particularly those relying on single facial features and needing help with large head movements. In response to the identified challenges, this study introduces an enhanced driver attention detection approach using an improved version of YOLOv5. The primary contributions of this research are outlined as follows: i) Introduce a streamlined framework for predicting 468 face landmarks and utilize these landmarks to calculate their aspect ratios.

ii) Integrating YOLOv5 backbone network to improve feature extraction, enabling accurate detection of small targets within the feature map.

iii) Incorporating the Swin Transformer module into YOLOv5’s feature fusion network, we enhance global perception and facilitate the fusion of feature maps of different sizes, replacing the Bottleneck module in the C3 module.

The research objective of the work is to advance driver drowsiness detection by refining the YOLOv5 object detection model to improve accuracy and effectiveness. The study aims to enhance the detection of driver drowsiness through three main innovations: firstly, by developing a streamlined framework to predict 468 facial landmarks and calculate their aspect ratios; secondly, by integrating an additional module into the YOLOv5 backbone network to improve feature extraction and enable precise detection of small targets; and thirdly, by incorporating the Swin Transformer module into the YOLOv5 feature fusion network to enhance global perception and facilitate the fusion of feature maps of different sizes. These advancements collectively aim to improve real-time monitoring and response to driver drowsiness, enhancing overall road safety.

Proposed methodology

Improved YOLO

A new technique for YOLOV5s object detection is presented, utilising a simplified spatial pyramid pooling fast (Sim SPPF) hybrid pooling method. This approach enhances feature fusion by combining shallow and deep feature information. It aims to boost machine vision’s capability to handle small target challenges and imbalanced sample issues. The enhanced YOLOV5s model enhances detection performance overall, outperforming other target detection models. The key improvements include: The improved YOLOV5s model incorporates the Sim SPPF structure, significantly enhancing feature extraction capabilities and object detection accuracy while speeding up calculations.

The attention mechanism channel attention (CA) is embedded in the head part to reinforce feature channels, ensuring the target detection model captures more crucial information.

The improved YOLOV5s model employs efficient intersection over union (EIOU) loss as the bounding box loss function, effectively addressing the issue of positive and negative sample imbalance and improving positioning accuracy.

The Sim SPPF and CA attention mechanisms are introduced based on the original YOLO backbone model. EIOU focal loss is also implemented to address the issue of difficult sample imbalance. Instead of using the aspect ratio from complete intersection over union (CIOU), the difference in aspect value is utilised. Sim SPPF is designed for industrial applications, emphasising detection and inference efficiency, and is more efficient than SPPF. Spatial pyramid pooling (SPP) can convert feature maps of varying scales into uniform scales. The CA attention mechanism learns an adaptive channel weight model, focusing on the most helpful channel information, resulting in high detection efficiency and optimal ablation.

Backbone network improvement

The backbone network of YOLOv5 utilises a Convolutional Neural Network (CNN) with a Cross Stage Partial Darknet (CSPDarknet) structure for object detection. Enhancing the feature extraction capacity involves integrating new modules following the SPPF structure.

The integration aims to precisely detect small objects by introducing three convolution operations and a new module after the feature extraction stage of the trunk network’s SPPF structure, with one branch undergoing convolution and the module while the other undergoes convolution alone. Afterwards, it goes through yet another convolution procedure following the concatenation operation. The aforementioned module is notable for its six separate convolution procedures. These operations yield different receptive fields, contributing to the enlargement of the receptive field through the stacking of convolution layers. This enables the extraction of features from various receptive fields, thereby enhancing the network’s feature extraction capabilities, particularly for detecting small-sized objects. The structure’s specifics are illustrated in Fig. 1, depicting the process of perceiving diverse feature information via multiscale receptive fields for optimal feature extraction. The module processes input feature information denoted as X and generates output information based on Eq. (1).

(1) Y=X1+X2+X3+X4+X5+X6

Figure 1 (A) Improvement of backbone network; (B) Modular structure.

The variables X1toX6 are defined as shown in Eq. (2):

(2) X1=cv1(X)X2=cv2(X)X3=cv3(X2)X4=cv4(X3)X5=cv5(X4)X6=cv6(X5)

The acronym cva is the convolutional operation, and + is the concatenation operation.

Swin transformer module

The Swin Transformer, introduced by Microsoft on March 25, 2021, presents a transformative approach to computer vision tasks by leveraging the Transformer framework. Swin Transformer mitigates increasing computational demands from larger visual entities and higher resolutions by employing sliding operations and a hierarchical design. Three fundamental modules compose its architecture: Foundational elements of the system include 1) Swin Transformer block, 2) Patch Merging, and 3) Patch Embedding. The Swin Transformer block, highlighted in Fig. 2, is the architecture’s central element, illustrating its crucial role within the system.

Figure 2 Architecture of Swin Transformer block.

In the Block1 stage, the input feature map Xi undergoes LayerNorm processing to derive LN (Xi). Next, the standardised feature map is calculated, which produces the layer’s intermediate output, Z^i, by combining the window attention operation with adding to the residual structure, as shown in Eq. (3). Following this step, further standardisation LN( Z^i)is applied. Finally, the fully connected layer is employed and integrated with the residual structure to produce Zi, as demonstrated in Eq. (4).

(3) Z^i=W−MSA(LN(Xi))+Xi

(4) Zi=MLP(LN(Xi))+Z^i

Block 1 of the Swin Transformer uses Window Multi-head Self-Attention (W-MSA), and Block 2 utilises Shifted Window Multi-head Self-Attention (SW-MSA), alternating these two mechanisms. Specifically, the transformer Blocks are structured in pairs, facilitating an organised and efficient processing flow.

C3 module incorporating Swin-Transformer

In this work, the YOLOv5 model integrates the C3 module, which has a bottleneck module and three convolutional modules. Unlike the standard modules, integrating Swin Transformer modules serves two primary purposes. Firstly, it reduces computational complexity and enhances network speed. Secondly, it enlarges the receptive field by utilising the sliding-window mechanism, improving the capacity to express features, especially for smaller targets. The adaptation enhances the model’s capability to capture global information during feature fusion or extraction, leading to the replacement of the traditional Bottleneck module within the C3 module by the Swin Transformer, as depicted in Fig. 3.

Figure 3 (A) Structure of C3 module; (B) structure of Swin module.

As the network structure deepens, the high-level feature map loses much of the target feature information, particularly for small targets in the image. To address this, we incorporate the idea of the Swin-Transformer into the feature fusion part, embedding it into the C3 module and replacing the four C3 models in the neck. By introducing discrete parameters of the Transformer and utilising the window self-attention module, we enhance the semantic information and feature representation of small targets. Figure 4 illustrates the C3STR with a Swin-Transformer.

Figure 4 Improved network structure diagram.

Swin-transformer features are learned by shifting the window, which enhances efficiency because self-attention is computed within the window. As long as the window size remains constant, the computational complexity of self-attention remains constant, making the total computational complexity a linear multiple of the image size. This reduces both the sequence length and computational complexity. The shifting operation enables interaction between neighbouring windows, establishing a cross-window connection between the upper and lower levels, thereby facilitating global modelling. The calculation process of the multi-headed self-attentive mechanism is as follows:

(5) Attention(Q,K,V)=SoftMax(QKTd+B)V

Attention denotes the attention mechanism, and SoftMax represents the normalisation function. Here, Q, K, and V are the query, key, and value matrices, respectively; d is the key dimension; and B is a smaller-sized bias matrix. The introduction of B can lead to significant improvements.

Improvement of neck network

The feature fusion network within YOLOv5 is inspired by the Feature Pyramid Network (FPN) structure, integrating a bottom-up path augmentation technique. This unification optimizes the utilization of both high-level and low-level feature data, thereby enhancing the model’s detection ability. Through upsampling in the FPN, the high-level features are merged with the low-level details, promoting a harmonious fusion of semantic richness and fine-grained information across various scales. The Path Aggregation Network (PAN) module performs bottom-up feature fusion, combining low-level and high-level features to produce three outputs for the final feature map, which is inputted into the detection layer.

A significant improvement in this research involves refining the network connection methodology. Unlike the original YOLO model, where the feature maps of sizes 80 × 80, 40 × 40, and 20 × 20 are merged in the feature fusion stage of the neck section; this study introduces a larger feature map (160 × 160) for more comprehensive information integration. Moreover, enhancements were implemented to better merge feature maps of varying sizes within the original network, thereby boosting the model’s ability to detect small objects. The adjusted structure of the network is illustrated in Fig. 4.

Loss function

Understanding the precise input and output values is crucial in model training to accurately calculate the loss function, quantifying the disparity between them. This study utilises the YOLOv5 model with a loss function consisting of three main components: The loss function comprises confidence loss, classification loss, and localisation loss. The classification loss and confidence loss are calculated using the Binary Cross-Entropy Loss method (BCE). In contrast, the loss of rectangular box localisation is assessed through the Complete Intersection over Union Loss technique (CIoU). Equations (6) and (7) represent the CIoU and Logarithmic Complete Intersection over Union (LCIoU), respectively. In contrast to prior techniques such as Generalized Intersection over Union Loss (GIoU Loss), IoU Loss, and Distance Intersection over Union Loss (DIoU Loss), the CIoU Loss method incorporates extra calculations with parameters like center point distance, aspect ratio, and overlap area. This refinement enhances the precision of the loss function by simultaneously factoring in multiple elements.

(6) CIoU=IoU−ρ2c2−αv

(7) LCIoU=1−IoU+ρ2c2+αv

Within the CIoU Loss method, ρ is the distance between the ground truth box and the centres of the prediction box, and c signifies the minimum rectangle’s length of the diagonal encompassing both boxes. Additionally, ν denotes the aspect ratio similarity between the two boxes, and α denotes the weight coefficient of ν, as outlined in Eqs. (8) and (9). These variables collectively contribute to the comprehensive calculation of the CIoU Loss, incorporating considerations such as center point distance, diagonal length, and aspect ratio similarity to refine the loss function.

(8) v=4π2(arctanωgthgt−arctanωh)2

(9) α=v1−IoU+v

Extraction of crucial facial landmarks and construction of a fatigue detection model

This section discusses the proposed driver drowsiness system illustrated in Fig. 5. While YOLOv5 is a powerful tool for object detection and localisation, facial 3D landmarks offer distinct advantages for tasks requiring detailed analysis of facial expressions and movements, such as fatigue detection in drivers. By combining 3D facial landmarks with YOLOv5 in driver fatigue detection, the proposed system benefits from the complementary strengths of both techniques, leading to more accurate and robust detection of fatigue-related cues and ultimately preventing accidents caused by drowsy driving. Initially, using 468 facial landmarks, the faces are detected, upon which improved YOLOv5 is employed to detect drowsiness. Algorithm 1 describes the step by step procedure of the prosed fatigue detection model.

Figure 5 Schematic representation of the proposed model.

Algorithm 1 Step 1: Extraction of 3D facial keypoints	
Input: Video Frame Image Iframe	
Output: 3D Facial Keypoints K3D	
       K3D = Extract 3D Facial Keypoints ( Iframe)	
Step 2: Eye-mouth aspect ratio calculation	
Input: Eye and Mouth Keypoints Keye, Kmouth	
Output: Eye Aspect Ratio (EAR), Mouth Aspect Ratio (MAR)	
       EAR, MAR = Calculate Aspect Ratio  (Keye, Kmouth)	
Step 3: Fatigued-driving detection model	
Input: Video Frames { I1,  I2,…, In}, Facial Keypoints K3D, EAR, MAR	
Output: Fatigue Detected (True for fatigued, False for normal)	
Fatigue Detected = Detect Fatigue ( Iframe, K3D, EAR, MAR)	

Extraction of 3D facial keypoints

Accurate coordinates of facial landmarks are essential to determine the eye and mouth aspect ratios. However, conventional methods like MTCNN’s five-point localisation offer limited insight, covering only the positions of essential facial landmarks such as the eyes, nose, and mouth corners. Consequently, they can only delineate facial contours without discerning signs of fatigue. Moreover, their reliance on a three-level cascaded network results in sluggish detection speeds. Similarly, the two dimensional 68 key points extracted by Dlib exhibit drawbacks such as feature loss and diminished real-time performance, particularly under significant head rotations. This study adopts the Attention Mesh architecture to address these limitations and ensure accurate, rapid, and stable extraction of facial key points with a focus on semantically significant regions. By predicting coordinates for 468 facial landmarks, Attention Mesh (Grishchenko et al., 2020) accurately determines the locations of vertices within a 3D facial mesh, presenting a streamlined approach for reliable facial landmark prediction.

Figure 6 illustrates a process involving a face extractor and an end-to-end feature extraction model. It starts by inputting the detection video frame into the system. The feature-extraction module separates the inputs into sub-models, independently predicting eye and mouth coordinates. Each sub-model adjusts its grid size to improve coverage. A normalization step aligns and resizes features for improved accuracy. This approach offers better facial keypoint localization accuracy and faster extraction than multi-stage methods.

Figure 6 Facial feature extraction using 3D facial landmarks.

Figure 7 visualizes the extraction of 3D facial key points. Key points for the left and right eyes are numbered 33, 133, 145, 154, 157, 159, 161, 163, 263, 362, 374, 381, 384, 386, 388, and 390. Key points for the outer contour of the mouth are represented by the indices 0, 17, 39, 61, 269, 181, 291, and 405.

Figure 7 Facial 468 key points of (A) open eyes, (B) open mouth, (C) closed eyes, (D) closed mouth, and (E) full face.

Where Iframe represents the current video frame image. (Keye, Kmouth) denote the extracted key points for eyes and mouth, respectively. K3D represents the 3D facial keypoints. EAR and MAR are the calculated eye and mouth aspect ratios, respectively. { I1, I2,…, In} denotes the set of video frames. Fatigue Detection indicates whether fatigue is detected or not.

Driver fatigue detection model with feature fusion

Eye-mouth aspect ratio and threshold determination for fatigue detection model with feature fusion

The eye aperture’s height fluctuates during blinking, dropping rapidly when the eyes close and gradually returning to a certain level. Conversely, during opening, it remains within a particular range. The study assesses the driver’s eye-opening and closing by calculating the eye-aspect ratio (EAR), as described in Soukupova & Cech (2016). Subsequently, a threshold value is determined based on this calculation.

In addition to detecting fatigue through changes in the driver’s eyes, yawning is another significant indicator of altered states. Yawning induces noticeable changes in the distance between the upper and lower lips and the distance from the left corner of the mouth, which briefly stabilises within a specific threshold range. This study introduces the mouth aspect ratio (MAR) derived from the EAR and determines the corresponding threshold to enhance the criteria for identifying fatigue-related conditions. To overcome the challenge of losing crucial inner mouth contour points due to variations in mouth features among drivers, this research calculates the mouth aspect ratio by isolating eight points from the external contour of the mouth. The formulas for calculating EAR and MAR are provided below.

(10) EARright=||Y384−Y381||+||Y386−Y374||+||Y388−Y390||3||X362−X263||

(11) EARleft=||Y161−Y163||+||Y159−Y145||+||Y157−Y154||3||X33−X133||

(12) EAR=EARleft+EARright2

(13) MAR=||Y39−Y181||+||Y0−Y17||+||Y269−Y405||3||X61−X291||

The equation includes the horizontal coordinates representation of four key points for each of the left and right eyes (X362, X263, X33, X133), as well as two key points for the mouth outline (X61, X291). Additionally, it denotes the vertical coordinates of twelve key points for both the left and right eyes (Y384, Y381, Y386, Y374, Y388, Y390, Y161, Y163, Y159, Y145, Y157, Y154), along with six key points for the mouth outline (Y39, Y181, Y0, Y17, Y269, Y405).

Figure 8 depicts an in-depth examination of a fatigued driver’s state to a normal state, with a specific focus on eye closure and yawning. This analysis utilizes randomly selected input video from the Yawning Detection Dataset (YawDD) (Abtahi et al., 2014). The graph illustrates the fluctuation of the mouth aspect ratio (MAR) as frame numbers progress. During typical instances of a closed-mouth state, the MAR remains relatively consistent within the (0.2–0.3) range. However, during yawning episodes, there is a sharp increase in the MAR, stabilizing around 1.1. Notably, when the MAR exceeds 0.65, it becomes evident that the driver is yawning. Thus, a yawning threshold of 0.65 is established (Gallup, Church & Pelegrino, 2016). Similarly, an eye-closure threshold of 0.02 is set, considering that the minimum eye aspect ratio (EAR) during closure approaches 0. Therefore, EAR < 0.02 indicates that the driver has closed their eyes.

Figure 8 EAR and MAR result analysis.

Figure 8 illustrates significant discrepancies in the counts of Fe (continuous frames with closed eyes) and Fm (continuous frames with yawning) between fatigued and normal driving states. Research indicates that individuals typically yawn for about 6.5 s, equivalent to approximately 150 frames. Evaluating a driver’s fatigue status involves calculating the number of consecutive eye-closed and yawning mouth frames using the following formula.

(14) Fe=Fej−Fei

(15) Fm=Fmj−Fmi

In the above equation, Fei, Fej, Fmi, and Fmj denote the start and end frames for eyes closing and yawning mouth.

PERCLOS and threshold determination

PERCLOS, which stands for Percentage of Eyelid Closure over the Pupil over Time, is crucial in assessing driver fatigue. Adhering to the P80 standard, a blink ratio below 0.2 indicates complete eye closure, while a ratio surpassing 0.8 indicates whole eye-opening. Exceeding a predefined threshold suggests potential driver fatigue. To enhance fatigue detection, new metrics such as accuracy and percentage of yawning in unit of time is introduced. Within a designated unit cycle frame, F0, PERCLOS scores for the eyes Peyes and mouth Pmouth are calculated based on the total number of frames showing eye closure and yawning.

(16) Peyes=∑FstartFend⁡(Fej−Fei)F0

(17) Pmouth=∑FstartFend⁡(Fmj−Fmi)F0

The equation incorporates Fstart and Fend, representing the initial and final frames within a designated unit cycle frame. An experiment utilised a video dataset to establish thresholds for eye and mouth fatigue based on EAR and MAR criteria. Continuous eye closure and yawning frames within a unit cycle were counted to compute the resulting PERCLOS score. Leveraging these indicators, a model for fatigued driving detection was devised. Analysis indicated that within a 150-frame unit cycle (F0 = 150 frames), if the PERCLOS score for the driver’s eyes and mouth remains above 0.15, if continuous eye closure frames exceed 20, or if continuous yawning frames surpass 30, the driver is identified as exhibiting fatigued driving behaviour; otherwise, they are classified as in a normal driving state.

(18) {F0=150Peyes=Pmouth≥0.15Fe≥20Fm≥30

Figure 9 outlines a fatigued-driving detection procedure designed to tackle challenges such as reduced accuracy and keypoint loss due to variations in head pose. This study presents a comprehensive approach to monitoring changes in drivers’ blinking and yawning patterns by integrating the extraction of three-dimensional key points, calculating the eye-mouth aspect ratio, and feature classification detection. The detection process involves assessing whether the driver’s mouth is open (o_mouth) and if the mouth ratio exceeds a predetermined threshold, indicating yawning. Conversely, if the driver’s eyes are closed (c_eyes) or the eye ratio falls below the specified threshold, it suggests blinking. These criteria are expressed mathematically as follows:

(19) {o_mouthΛ(MAR>0.65)c_eyesV(EAR<0.02)

Figure 9 Driver drowsiness detection process.

A holistic approach to identifying fatigued driving involves analysing eye and mouth features. This method evaluates the driver’s status by analysing the duration of closed-eye instances, consecutive yawning occurrences, and predetermined fatigue thresholds for eye and mouth behaviours. The output for this model is either normal driving or fatigued driving.

Experimental framework

Dataset description and experimental conditions

The dataset encompasses 8,021 images sourced from diverse datasets, including well-known public ones like YawDD, CEW, DrivFace, and Drozy. The dataset includes drivers from various demographics, including genders and ethnicities. Some drivers wear glasses, while others do not, adding to the dataset’s diversity. It captures normal and fatigued driving instances, including scenarios where drivers are engaged in conversation or remain silent. The dataset includes curated videos showcasing real-world driving situations under different lighting conditions, both day and night.

To prepare the dataset for training, video data from YawDD and self-compiled sources undergo several preprocessing steps, including mirroring, rotation, and cropping, resulting in images captured at intervals of 20 frames. Additionally, supplementary datasets such as CEW, DrivFace, and Drozy are incorporated to augment diversity and account for scene variations. This augmentation approach strengthens the dataset’s resilience and its ability to generalize. The dataset annotation process utilizes the Labeling Python library, creating 8,021 face bounding-box labels. The dataset includes 3,579 bounding-box labels for open mouth (o_mouth), 4,634 for open eyes (o_eyes), 4,310 for closed mouth (c_mouth), and 3,166 for closed eyes (c_eyes), totaling 23,710 facial feature bounding-box labels across different categories. Following annotation, the dataset is divided into training and validation sets using an 8:2 ratio to aid in model training and evaluation.

During the model training phase, the training schedule spans 150 epochs, with images resized to a dimension of 640 and processed in batches of 16. The rect matrix method is implemented to reduce unnecessary padding in image preprocessing, enhancing training efficiency by reducing memory consumption during training and improving inference times.

The system’s Central Processing Unit is an Intel Xeon Platinum 8255C, while its Graphics Processing Unit is an NVIDIA RTX 3080 with 12 GB of memory. The system’s memory capacity is 48 GB. For deep learning tasks, the system employs PyTorch version 1.12.0 as its Deep Learning Framework (DLF) and is programmed using Python 3.9. GPU acceleration is facilitated by CUDA version 11.5. This combination of hardware and software components provides a powerful platform for conducting deep learning experiments and model training.

Performance metrics

The study evaluates upgraded face-detection algorithms using metrics including average precision (AP), mean average precision (mAP), floating-point operations (FLOPs), parameters (Params), and model size (Size). AP and mAP measure the algorithm’s accuracy in predicting targets, indicating its ability to recognise faces across various categories and overall effectiveness. Lower FLOPs indicate reduced computational complexity, while smaller Params and Size values signify a lighter model. The formulas for calculating average precision (AP) and mean average precision (mAP) are below.

(20) P=TPTP+FP

(21) R=TPTP+FN

(22) AP=∫01P(R)dR

(23) mAP=∑i=15⁡APi5

The equation involves variables where P stands for precision, and R signifies recall. When assessing the model’s performance, TP indicates correctly predicted faces within each category, while FP represents incorrectly predicted faces within each category. FN represents the number of faces within each category the model failed to predict. AP represents the average precision of predicting each category within the face dataset, with the value 5 corresponding to the five feature categories of faces classified.

Experimental results and discussion

Table 2 compares various models using the same custom dataset. The enhanced YOLOv5 model outperforms prevalent one-stage detection algorithms like YOLOv4 and YOLOv7 regarding average precision. It exhibits an improvement of 1.1% over YOLOv4 and 0.8% over YOLOv7. Additionally, it exhibits significant enhancements in detection speed, maintaining a satisfactory recognition rate despite a slight decrease in frames per second (FPS) to 55 f/s. Moreover, the enhanced YOLOv5 model boasts a smaller size of only 17.5 MB, making it more cost-effective to deploy and easier to integrate into various applications, including automotive systems.

Table 2 Performance comparison of various networks.

Model	mAP/%	FPS/(f/s)	Size (MB)	
YOLO_v4	85.22	24.4	254	
YOLO_v5s	83.91	63.3	13	
YOLO_v7	85.53	49.6	71	
Improved YOLOv5	86.30	55.8	17	

The diverse enhancement methods were trained and evaluated on the custom dataset, and their impacts on the base model are summarised in Table 3. Integrating the Swin Transformer module expanded the model’s receptive field, improving its capability to capture global information and enhancing detection accuracy. Despite a slight decrease in detection speed due to adjustments in the backbone network and connection method, there was a notable enhancement in the accuracy of detecting smaller objects.

Table 3 Ablation study.

Swin Transformer	Improvement of backbone	Improvement of neck	mAP/%	FPS/(f/s)	Size (MB)	
–	–	–	82.91	64	13.8	
✓	–	–	84.23	58	13.7	
–	✓	✓	84.65	59	17.3	
✓	✓	✓	86.32	56	17.4	

Figure 10 displays the enhanced classification detection outcomes, illustrating the benefits of the refined algorithm through ablation experiments. These illustrations demonstrate the algorithm’s effectiveness in predicting facial features across different categories. Particularly, integrating Exponential Moving Average (EMA) enhances the extraction of comprehensive facial feature details, providing precise localisation data for categories with smaller targets and subtle features. This enhancement significantly boosts attention on feature regions, particularly in the c_mouth and c_eyes categories. Importantly, there’s a notable improvement in the recall and mean average precision of c_eyes, with increases of 9.2% and 2.3%, respectively.

Figure 10 Results for classification before/after improvement: (A) YOLOv5s (B) Ours.

The upgraded algorithm’s effectiveness in classification detection was assessed using a subset of 50,000 facial images randomly selected from the CelebA dataset, covering various facial features and environmental conditions. Evaluation centered on classification recognition accuracy, revealing promising results detailed in Table 4. The algorithm achieved an impressive recognition rate of 98.6% for facial feature target regions by leveraging feature fusion enhancements from Maxpool-Cross scale Feature Aggregation Module (M-CFAM) and Lightweight-Contextual Information Fusion Module (L-CIFM). Notably, it demonstrated near-perfect recognition rates for the face and o_eyes categories, showcasing precision in identifying these features. However, higher rates of false detections were observed for the c_eyes and c_mouth categories, indicating areas for potential improvement in classification decisions. Nevertheless, the algorithm maintained an average recognition rate exceeding 97%, demonstrating commendable overall performance during evaluation.

Table 4 Results of the improved algorithm’s detection.

Classes	Total count	Correctly identified	Accuracy	
O_Eyes	28,870	28,690	0.993	
C_ Eyes	19,617	18,734	0.954	
O_Mouth	22,345	22,016	0.985	
C_ Mouth	26,994	26,380	0.977	
Face	50,000	49,850	0.997	
Comprehensive	147,549	145,379	0.985	

The results presented in Table 5 demonstrate a consistent improvement in mean average precision (mAP) across all categories. The recognition precision (P) for the face, o_mouth, o_eyes, and c_mouth categories remains comparable to that of the baseline model. Systematic optimisation and enhancements to each component of the baseline model enable accurate extraction and prediction of facial features and categories while minimising system overhead. These results highlight the effectiveness of the upgraded algorithm.

Table 5 Performance of the proposed model on custom dataset to test the effects of eyeglasses during daytime and night time.

	Day time	Night time	
Scenario	Non-drowsy (%)	Drowsy (%)	Scenario	Non-drowsy (%)	
With eyeglasses	94.3	95.9	With eyeglasses	94.4	
Without eyeglasses	96.6	97.8	Without eyeglasses	96.7	
Average	95.8	96.6	Average	95.	

The Single Shot Detector (SSD) model is fast with medium accuracy and high complexity, making it ideal for general use and multi-scale detection. YOLOv3-Tiny and YOLOv4-Tiny prioritise very fast speeds with low complexity, with YOLOv4-Tiny offering better accuracy, making them suitable for real-time low-power applications and small object detection. YOLOv5n is ultra-light and very fast, performing better than YOLOv4-Tiny, making it ideal for ultra-efficient mobile devices, while YOLOv5s strikes a balance between speed and accuracy for general use. YOLOv7-Tiny offers fast speed with high accuracy and light complexity, suitable for speed- and accuracy-critical tasks. YOLOv8s is fast and offers the highest accuracy among these models, excelling at small object detection in modern use cases. Compared with all other YOLO versions, the proposed model outperforms traditional YOLO models by combining YOLO’s real-time detection speed with the Swin Transformer’s superior feature extraction and attention mechanisms. The Swin Transformer excels at capturing long-range dependencies and handling varying object scales, significantly improving accuracy, particularly for small or occluded objects in complex scenes.

Through an extensive performance evaluation compared to existing mainstream algorithms, our proposed algorithm distinguishes itself with a notably compact model size of merely 6.3 MB, considerably smaller than several counterparts such as SSD, YOLOv3-Tiny, YOLOv4-Tiny, YOLO_v5n and YOLOv7-Tiny. Despite a slightly higher volume of floating-point operations compared to some models, our algorithm exhibits notable improvements in mean average precision (mAP), ensuring robust performance in predicting facial-feature categories while maintaining its lightweight nature. While it may lag behind the YOLOv8 series in mAP and detection performance, its significant advantage lies in its lightweight design and superior overall performance. This highlights the algorithm’s capacity to attain significant robustness while utilising minimal computational resources, fulfilling the portability needs of embedded mobile devices and affirming its advancement effectively.

This investigation integrates the trained weight file from the enhanced algorithm into the evaluation model for fatigued-driving detection, enabling multi-index and multi-feature fusion for comprehensive detection. To evaluate the model’s accuracy, 135 segments averaging 560 frames each were extracted from the YawDD video dataset for testing purposes. These segments cover scenarios of normal and fatigued driving, including instances with and without drivers wearing glasses, encompassing different genders and driving conditions. The evaluation metric relies on recognition accuracy derived from instance detection. The effectiveness of the fatigued-driving detection model is compared to previous studies (Ji et al., 2019; Liu et al., 2019), which utilise the MTCNN five-point positioning method. The comparative analysis results in Table 6 provide a detailed performance assessment for further scrutiny.

Table 6 Validation results of mainstream algorithms on datasets.

Method/Model	Params [M]	mAP	FLOPs/G	Size [MB]	
SDD	18.42	0.774	28.39	104.1	
YOLO_v3-Tiny	7.62	0.876	13.2	52.3	
YOLO_v4-Tiny	7.05	0.881	3.4	21.5	
YOLO_v7-Tiny	5.03	0.956	13.2	13.4	
YOLO_v5n	1.67	0.928	4.2	3.7	
YOLO_v5s	6.03	0.949	15.7	14.2	
YOLO_v8s	10.30	0.977	28.7	24.6	
Ours [YOLO-Swin]	2.94	0.957	5.8	6.2	

Table 7 highlights deficiencies in fatigued-driving detection in Ji et al. (2019), particularly in discerning normal states, eye changes, and yawning. Notably, fatigue recognition during yawning is notably low at 23.1% compared to 98.1% in Liu et al. (2019) and 96.7% in our study. Utilizing combined eye and mouth features, our model achieves an overall judgment accuracy of 96.3%, with recognition accuracy for fatigued driving and normal driving reaching 93.3% and 97.8%, respectively. While slightly trailing Liu et al.’s (2019) 97% comprehensive performance, maintaining similar conditions results in 97.5% overall accuracy, demonstrating our model’s robustness, as depicted in Fig. 11. Though lacking validation in real driving scenarios, our model outperforms in detection and discrimination. Further enhancements are needed to strengthen classification discrimination ability, particularly in inconspicuous driver states.

Table 7 Results comparison and fatigue detection.

Method/Model	Category	Quantity/Size	Correct identification	Error	Accuracy (%)	
Ours	Normal	50	50	–	1	
Yawning fatigue	35	33	Normal	0.94	
Eye fatigue	14	13	Normal	0.92	
Comprehensive	140	135	–	0.96	
Ji et al. (2019)	Normal	110	67	Tired eyes + yawning	0.60	
Yawning fatigue	82	19	Normal	0.23	
Eye fatigue	33	23	Normal	0.69	
Comprehensive	325	206	–	0.63	
Liu et al. (2019)	Normal	10,291	9,643	Tired yawning	0.93	
Yawning fatigue	21,643	21,234	Normal	0.98	
Eye fatigue	–	–	–	–	
Comprehensive	44,838	43,491	–	0.97	

Figure 11 Results to display drowsy or awake with or without glasses.

Conclusion and future work

In this proposed work, fatigue detection relies on calculating the aspect ratios of the eyes and mouth, while distracted behavior detection involves enhancing the YOLOv5 model with multiple convolution operations in its backbone network to widen receptive fields and improve feature extraction. Furthermore, replacing the Bottleneck module with the Swin Transformer module in the C3 module of the feature fusion network enhances global information awareness, while additional improvements in network connection modes further enhance feature fusion capability. To address computational complexity, parameter demands, and limitations in facial feature keypoint extraction in fatigued-driving detection models, we propose a novel lightweight and real-time fatigued-driving detection model. The model, built on an improved YOLOv5s architecture and employing the Attention Mesh 3D key point extraction method, showcases superior mean average precision (mAP) compared to the baseline YOLOv5 model and surpasses the newer YOLOv7 model in terms of smaller model size and higher detection accuracy. Future research will prioritize optimizing the real-time performance of the network model for further efficacy.

Additional Information and Declarations

Competing Interests

Author Contributions

Data Availability

The authors declare that they have no competing interests.

Mohan Arava conceived and designed the experiments, performed the experiments, analyzed the data, performed the computation work, prepared figures and/or tables, and approved the final draft.

Divya Meena Sundaram conceived and designed the experiments, prepared figures and/or tables, authored or reviewed drafts of the article, and approved the final draft.

The following information was supplied regarding data availability:

The YawDD dataset is available at GitHub: https://qualinet.github.io/databases/video/yawdd_a_yawning_detection_dataset.

The CEW dataset is available at: http://parnec.nuaa.edu.cn/_upload/tpl/02/db/731/template731/pages/xtan/ClosedEyeDatabases.html

DOI: 10.1016/j.patcog.2014.03.024

The DrivFace dataset is available at GitHub and UCI:

- https://github.com/adrshm91/DrivFace

- Hernndez-Sabat, A., Lpez, A., & Diaz-Chito, K. (2016). DrivFace [Dataset]. UCI Machine Learning Repository. https://doi.org/10.24432/C5XC7Q.

The DROZY dataset is available at: http://www.drozy.ulg.ac.be.

The code is available at GitHub: https://github.com/DivyaMeenaSundaram/Driver-drowsiness-detectionn.

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
