# Peer review of "Integrating lightweight YOLOv5s and facial 3D keypoints for enhanced fatigued-driving detection"

_PeerJ Computer Science, doi:10.7717/peerj-cs.2447_

## Round 0.1 · original submission · Major Revisions

· Academic Editor

Major Revisions

Based on the reviewers' feedback, we have decided that your paper requires a major revision before it can be considered for publication.

In what follows some comments raised by the editor that need to be addressed for improving the overall quality of the work.

AE Comments:
Grammar and Readability: The paper requires a revision of the English, and some parts are unclear.

Specific Comments: My specific comments concerning this manuscript are:

- The abstract does not highlight the specifics of the research or findings but contains too much background information. Some details of the research would be nice for example numbers addressing the sample, data, percentage improvement, etc.. Remove some of the background material and add some details of the research. Moreover, it is good to provide some specifics (e.g., sample size, dataset size, numbers from results, etc.).

- Although a novel combination might be allowed, it is necessary to highlight the contribution of such a combination from both methodological and empirical perspectives. Also, it is required to provide technical details of the proposed methods as much as possible and in-depth explanations of method selections.

- The innovation of the paper seems limited. The proposed method is a straightforward combination of existing techniques, which makes it less innovative. Also, more details about the proposed method should be provided.

- I suggest adding a clear research objective or research questions in the introduction section and specify what the main research problem or hypothesis is addressed

- I recommend proofreading to make reading smoother.

- There are several papers that have addressed similar problems, but it is necessary to further highlight the novelty between the proposed study and the related literature.

- Starting from the previous works, I suggest introducing a table to summarize the most recent works and to highlight the novelty of the proposed work.

- Please add more recent references. Certainly, there has been more recent (within the last two years) research on this topic published in information science and/or computer science outlets. An academic search on the topic (using keywords from the manuscript’s title) shows that there is recent work in this area. Therefore, authors must update their literature review.

- There needs to be an explicit research objective(s) and/or research question(s) stated, preferably as a separate section. This helps readers find out what the research is trying to address.

- The reference list needs tidying up, as there are references missing items or formatting issues. Please be consistent with the formatting and use some standard formatting style.

- The evaluation is weak. Please consider using a more convincing way to evaluate the proposed method, e.g., using more datasets.

- The discussion of the results does not highlight the strengths and weaknesses of the proposed approach.

Concluding Remarks:

I think that the paper could be improved with the considerations I reported in the review, but this version is not ready for publication.
* * *
Moreover, we recommend that you carefully proofread your manuscript to resolve any typographical errors and other mistakes. A meticulous review of your paper will contribute significantly to its clarity and professionalism.

Reviewer 1 ·

Basic reporting

The manuscript entitled “Integrating lightweight YOLOv5s and facial 3D keypoints for enhanced fatigued-driving detection” has been investigated in detail. This paper aims to address the issue of drowsy driving by proposing an enhanced YOLOv5 algorithm integrated with Swin Transformer modules for real-time detection of driver fatigue. While the study presents some innovative approaches, several critical issues need to be addressed to improve the clarity, robustness, and applicability of the research.
1) The paper claims to introduce an "advanced YOLOv5 model integrated with Swin Transformer modules" but lacks detailed explanation on how these components specifically enhance the model’s performance. Provide a clear comparison with existing models to highlight the novelty of your approach.
2) The claim of a 2.4% enhancement in mean average precision (mAP) is significant but needs more context. Compare this improvement to more recent or diverse baseline models to validate the competitiveness of your approach.

Experimental design

3) The methodology section is vague about the specific modifications made to the YOLOv5 architecture. Detailed descriptions of the Swin Transformer modules, feature fusion network, and refined backbone network are necessary for reproducibility.
4) Explain how the threshold values for eye aspect ratio and mouth aspect ratio were determined. Were these values empirically derived, and how do they generalize across different populations or driving conditions?
5) Specify which benchmark datasets were used for the experiments. Include details on the dataset size, diversity, and how representative they are of real-world driving conditions.
6) Besides mAP, include other evaluation metrics such as precision, recall, F1-score, and computational efficiency (e.g., inference time) to provide a holistic view of the model’s performance.

Validity of the findings

7) Discuss the computational requirements and efficiency of the proposed model. Is the model capable of running in real-time on standard hardware used in vehicles?
8) Evaluate the model’s performance under different driving conditions (e.g., night vs. day, varying lighting conditions) and its robustness to different driver demographics (e.g., age, gender, ethnicity).

Additional comments

The manuscript entitled “Integrating lightweight YOLOv5s and facial 3D keypoints for enhanced fatigued-driving detection” has been investigated in detail. This paper aims to address the issue of drowsy driving by proposing an enhanced YOLOv5 algorithm integrated with Swin Transformer modules for real-time detection of driver fatigue. While the study presents some innovative approaches, several critical issues need to be addressed to improve the clarity, robustness, and applicability of the research.
1) The paper claims to introduce an "advanced YOLOv5 model integrated with Swin Transformer modules" but lacks detailed explanation on how these components specifically enhance the model’s performance. Provide a clear comparison with existing models to highlight the novelty of your approach.
2) The claim of a 2.4% enhancement in mean average precision (mAP) is significant but needs more context. Compare this improvement to more recent or diverse baseline models to validate the competitiveness of your approach.
3) The methodology section is vague about the specific modifications made to the YOLOv5 architecture. Detailed descriptions of the Swin Transformer modules, feature fusion network, and refined backbone network are necessary for reproducibility.
4) Explain how the threshold values for eye aspect ratio and mouth aspect ratio were determined. Were these values empirically derived, and how do they generalize across different populations or driving conditions?
5) Specify which benchmark datasets were used for the experiments. Include details on the dataset size, diversity, and how representative they are of real-world driving conditions.
6) Besides mAP, include other evaluation metrics such as precision, recall, F1-score, and computational efficiency (e.g., inference time) to provide a holistic view of the model’s performance.
7) Discuss the computational requirements and efficiency of the proposed model. Is the model capable of running in real-time on standard hardware used in vehicles?
8) Evaluate the model’s performance under different driving conditions (e.g., night vs. day, varying lighting conditions) and its robustness to different driver demographics (e.g., age, gender, ethnicity).
The paper presents an enhanced YOLOv5 algorithm for detecting driver drowsiness, incorporating advanced features like Swin Transformer modules and Attention Mesh 3D keypoints. While the proposed method shows promise, it requires significant revisions to improve the clarity of the methodology, robustness of experimental validation, and discussion on real-world applicability and ethical considerations. Detailed technical descriptions, comprehensive evaluations, and practical implementation discussions are essential to strengthen the paper.

Reviewer 2 ·

Basic reporting

1. Please define drowsiness and fatigue. Are they the same? Explain the relationship.
2. improve the resolution of the figures
3. Proofread the writing format. use punctuations', etc.

Experimental design

Section 3, paragraph 2, line 337 - what do you mean by self-compiled sources?

Figure 6 & 7 - please put picture from the datasets, to see that the data was taken during driving. Current pictures were in the lab/ office setup.

The author has made few comparisons with other YOLO models/ algorithms - please explain this in Section 3.2.

Validity of the findings

Section 4, line 374 - what is custom dataset?

The performance of the proposed methods was compared with other YOLO models: v4, v5s and v7. What is the difference between these models and the proposed one?

Also, about validations with other algorithms (Table 4). Should explain how they were conducted?

Additional comments

1. Abstract
- How was the real-time fatigue-driving detection model conducted? The author used public datasets and didn't mention about the real-time testing.
- Line 19: what are the conventional methods? The author should compare the performance of the proposed method with the conventional methods.
- How was the 2.4% enhancement achieved? Please explain explicitly in Results section.
- The Abstract should be improved as it does not represent the overall work clearly

---

## Round 0.2 · accepted · Accept

· Academic Editor

Accept

I hope this message finds you well. After carefully reviewing the revisions you have made in response to the reviewers' comments, I am pleased to inform you that your manuscript has been accepted for publication in PeerJ Computer Science.

Your efforts to address the reviewers’ suggestions have significantly improved the quality and clarity of the manuscript. The changes you implemented have successfully resolved the concerns raised, and the content now meets the high standards of the journal.

Thank you for your commitment to enhancing the paper. I look forward to seeing the final published version.

Reviewer 1 ·

Basic reporting

It is acceptable in the present form.

Experimental design

It is acceptable in the present form.

Validity of the findings

It is acceptable in the present form.